# Work–Life Balance and Teleworking: Lessons Learned during the Pandemic on Gender Role Transformation and Self-Reported Well-Being

**DOI:** 10.3390/ijerph19148468

**Published:** 2022-07-11

**Authors:** Ana M. González Ramos, José María García-de-Diego

**Affiliations:** 1Institute of Advanced Social Studies, Spanish National Research Council (IESA-CSIC), 14004 Cordoba, Spain; 2Sociology Department, University of Granada, 18011 Granada, Spain; jmgdediego@ugr.es

**Keywords:** well-being, teleworking, gender regime, COVID-19 pandemic, gender role distribution, feelings about work–life balance, life satisfaction

## Abstract

Lockdown during COVID-19 forced the emergence of a new scenario, with men and women teleworkers spending all their time at home. The purpose of this study is to address whether this situation has triggered a transformation in gender roles and self-reported well-being, comparing the responses of male and female respondents to the EUROFOUND April to July 2020 survey. The analysis addresses cultural differences across European regions related to diverse gender regimes, employment status, and the possibility of teleworking. It explores male and female well-being through life satisfaction, the distance between happiness and life satisfaction, and rates their feelings about work–life balance. Findings on life satisfaction display a low transformation of social roles, with women still worrying about work–life balance, while men were more affected by the health crisis. Men self-report high life satisfaction across Europe compared to women, although unexpectedly, female freelancers in Northern and Southern European had a higher life satisfaction ratio than men. Both men and women teleworkers reported difficulties with managing work–life balance at home, despite women handling core care and household tasks. These findings suggest that women would have received more support from men, as they worked harder and longer hours during the lockdown, despite their weak position in the labor market. This would seem to be a propitious setting for men to have assumed more responsibility at home, resulting in a more equal distribution of roles at home.

## 1. Introduction

Telework has been a beacon of hope for gender equality policy since it should potentially improve women’s work–life balance [1]. However, in the late twentieth century, scholars were already issuing warnings about home-based teleworking representing a risk factor since workplaces remain inequal and gender regimens remained biased [2,3,4,5]. Since then, distance work has become progressively more widespread as the digitalization of the economy continues. The “domestication” of ICT [6] and globalization have substantially transformed work toward a kind of “work arrangement where workers work remotely, away from an employer’s premises or fixed location, using digital technologies such as networks, laptops, mobile phones and the internet” [7], (p. 1). Telework has expanded to embrace diverse work types, which has deeply transformed labor conditions as well as workers’ lifestyles [8,9]. Technological and social transformations do not seem enough to erase gender bias [10]. Until today, men and women have had entrenched unequal conditions, with gender-based roles and tasks present both in the domestic and public domains, as described below in this paper.

Androcentrism underlies female discrimination in the labor market and domestic domain, adding an ever-greater accumulation of social roles to accomplish, and public policy focused on erasing differences (on gender issues and social classes). The emergence of the concept of a work–life balance was a foregone conclusion, addressing women who work double shifts but, over time, has benefited both men and women, with the aim of equitable distribution of tasks and enjoyment of private time. The gender gap is still present in work culture [11]. There is evidence such as the gender pay gap, the glass ceiling in high positions at organizations, segregation of jobs by gender, and harassment. The discussion is still open about whether women are career-centric, arguing that they are less motivated to pursue professional goals, pushed or pulled by family [12,13]. The incorporation of women into the labor market is difficult and irregular, depending on their cultures, countries, and lifestyles [14]. Northern and Southern European countries are examples of different gender regimes and distribution of teleworking and values referring to work and lifestyles. Whereas Northern countries were historically leaders in advanced gender policy, teleworking is more widespread and time organization is tight, making it difficult to respect employee-friendly schedules and personal issues, Southern countries have worse labor conditions, a higher degree of presenteeism culture at workplaces, and deeper gender-role prescriptions.

In addition, the work ethic of neoliberalism has been deeply ingrained in workers’ lives, constantly pursuing achieving their goals and accomplishments at increasingly faster speeds and incessant multi-tasking. Social acceleration [13] affects workers’ life satisfaction, particularly women workers (who have the mandate of being good mothers, workers, colleagues, and leaders), juggling multiple social roles and time regime constrictions [15].

Additionally, the COVID-19 pandemic has strengthened digitalization as the result of social distancing and lockdowns [16]. For the first time, men and women were at home undertaking work and family duties at the same time. However, who was responsible for childcare, household chores, and nursing, traditional roles of women? Did men take the opportunity to assume care duties? If men did indeed take on more roles, did men feel stressed and worried about the “broken social norms”? Conversely, did women feel hopeful that there might finally be changes in these social norms? Did men or women feel their well-being deteriorated? We asked these questions to learn about the (un)certain transformation of gender-role balance and whether there was an unraveling of social roles, at least under the special crisis circumstances, and if so, what we should learn from a gender perspective.

### Are Changes Possible?

Although social changes involve all men and women, most gender studies center on women as workers or how they are treated by institutions on the path towards equality, emphasizing the role of women in this social process. When scholars compare gender regimes and what kinds of transitions from domestic to public realms are transforming gender relations, what they are looking for is how switching roles affects women [17,18]. They are the social actors who are doing and undoing social changes, and they are also the vulnerable group in need of the public policy. This framework is rooted in the evidence that social conditions are primarily what push women into male circumstances. Over time, women are tackling spaces that were traditionally male domains, while men remain standing in the same place. All risks are taken by women, who move from one social role to another, juggling domestic and employment tasks, accomplishing demands with accelerated timelines, and increasing demands on well-being. Time distribution polls still report women spending more hours on childcare and housework, while men keep privileges as breadwinners and work-centric public domain actors [19,20]. No social forces are pushing men into the domestic realm, none so strongly as women are pushed by employment and social dynamics. Teleworking has different connotations for men and women as a result of traditional gender roles [8]. Whereas male teleworkers are highly mobile employees and self-employed, more women than men are teleworkers at home with poor conditions and low-ranking jobs in the company. In the same study, findings display quite similar satisfaction with work–life balance with telework, although, in the group of highly mobile workers and occasional teleworkers, women are more likely to report that working hours fit well or very well with their lifestyle than men. This suggests that men hold more critical opinions than women of their lifestyles and that women are more satisfied with their work schedules when teleworking. Although patriarchy characterizes all countries, the cultural differences across regions play a role in gender equality, work–life balance, and well-being [21,22,23]. Generally, Northern European countries present the lowest levels of work–life conflict because of gender and social inequality and the regulation of labor relations. Meanwhile, Southern countries reported more domestic division and work–life conflicts. The work–life conflict would be associated with poor self-reported health for men and women, although there are diverse interfering factors, such as the tradition of public childcare support for families, women’s part-time regimes, dual-income couples, willingness for professional development, public policy and benefits, and so forth. 

Thus, cultural differences affect the self-reported well-being of men and women regarding the work–life conflict. Mensah and Adjei [23] found the smallest association between poor self-reported health indicators for both men and women in Nordic and Southern welfare states. Conversely, Liberal countries (Anglo-Saxon countries with strong male breadwinner tradition, minimal social policies, and poorly-regulated labor market), Conservative countries (also breadwinner model but strong labor laws), and Central Eastern concentrate the largest association between poor work–life balance and poor self-reported health. They explain these findings as a combination of factors related to a strong male breadwinner tradition, extension of social policy and provision of family-care benefits, employment regulations, and union protection. 

Lockdowns threw off social transformation for everyone, although the main change would be on the men’s side, who traditionally have rarely settled in the domestic arena, and the pandemic led to them spending more time at home. Has this new scenario brought changes in gender relations? What changes have affected men locked down at home and exposed daily to domestic issues? Did they do chores and take care of children? What roles and tasks have replaced the time they once spent on work relationships and public affairs? Rubery and Tavora collected evidence from cross-sectional surveys during the pandemic that suggests a shrinking gap between mothers and fathers in childcare hours spent [24]. However, the authors are cautious with these findings and warned that work–life balance was undertaken differently between the health crisis and “normal” times. Conversely, data show information about women feeling resistant to assuming caretaker roles during the pandemic as professional and supported agents of family care and household well-being [16,25]. 

Differences in cultural and political traditions also influenced policy responses during COVID-19 and the populations’ risk perception [24]. In that sense, have men and women from European regions revealed different roles and reactions with regards to work–life balance as the result of their exposure to care roles and working at home? Have they perceived the risk of well-being in the same way, or did they experience different worries? What kind of risk determinants felt related to juggling family and work chores? The question was posed about what men and women have felt about teleworking and handling family and jobs in the same place during the pandemic, and more importantly, what mattered to men? What did they do, and how did they feel spending more time at home exposed to care roles in domestic settings? 

Eurofound [25] has reported low female resilience more often than male respondents as a result of their vulnerable situation during the pandemic (quality of employment, economic sector, poor labor conditions, etc.). Results were formulated on the basis of two questions in the survey rated on five levels of satisfaction: “I find it difficult to deal with important problems that come up in my life” and “When things go wrong in my life, it generally takes me a long time to get back to normal.” This relevant insight is related to toughness to withstand adverse conditions but disregards what they felt when facing daily work–life balance experiences by gender, the focus of this work.

## 2. Materials and Methods

From April to July 2020, Eurofound launched a two-round online survey entitled “Living, Working and COVID-19,” consisting of 11,575 total panel questionnaires across EU-27 countries. The crisis contextualization may also have affected the number and motivation of respondents at home to endorse the surveys’ claims. Responses were gathered by uncontrolled convenience sampling through public dissemination among Eurofound contacts and stakeholders, social media advertising, and targeting hard-to-reach groups [26]. Therefore, the sample may have a self-selection recruitment bias. The dataset includes two rounds, where the fieldwork for Round 1 took place between 9 April and 11 June 2020 and Round 2 between 22 June and 27 July 2020 [26]. 

The analysis of this work aims to compare how men and women felt during the COVID-19 lockdown in various contexts depending on their gender regimes and work–life balance experiences. We were particularly interested in learning more about male teleworkers because we understand they were in a brand-new setting that may have led to their doing more in domestic realms. If they did, we want to know what risks they perceived to their lifestyle and well-being. We are aware that the population has preliminary cultural differences related to teleworking and gender regime—even more so in this period—with mortality rates and health-risk perceptions that may have swayed their thoughts and self-reported feelings during this period. In summary, we first needed to determine what combination of demographic factors best classified all the different individuals (male and female workers teleworking or not in different countries and regions). Table 1 below details the characteristics of the study sample for the purpose of this analysis.

A classification tree analysis was conducted to learn which factors are based on the categorical variable for life satisfaction. This technique is intended to divide the populations or samples under study, following a descending sequential process, for the identification of homogeneous subgroups in a variable of interest to the researcher (dependent variable, response, or criterion), taking into account the characteristics of the subjects studied in a selection of the independent variables, predictors, prognosticators or most important explanations for them. This data mining technique aims to identify homogeneous subgroups of the population with respect to certain characteristics and which independent variables show the greatest differences with regard to life satisfaction scores. A dendrogram was obtained from the socio-demographic variables on the population by the greatest weight in the responses for the constructed indicators: gender, work at home, and European countries group. Population subgroups were formed by the importance of the relationships between variables, and a further characteristic was added according to its degree of importance based on statistically significant differences. The process classifies individuals in several groups according to the most important factors determined by responses with significant differences between variables, differentiating which factors are different among homogenous subgroups of the population.

To learn more about the feelings of men and women during this period of time, the analysis addresses the distance between happiness and life satisfaction, an indicator proposed by Ruut Veenhoven [27,28] to develop a comprehensive indicator of the quality of life between nations. To analyze this distance, an ANOVA has been used, which is one of the most widely used statistical techniques to compare groups of measures, which is normally used to establish similarities and differences between three or more different groups. Through ANOVA, an analysis is established to comparatively evaluate results in different classifications or groups. In this way, it is possible to calculate if the mean values are the same in the different groups studied. Happiness is considered a subjective enjoyment of life, whereas life satisfaction is related to how much the person likes their life. As both variables correlate [28], subtracting happiness from life satisfaction, the measure would indicate a subjectivation of respondents’ quality of life. 

### Description of Variables

Northern, Southern, Eastern, and Western European countries follow the United Nations’ classification (according to: https://unstats.un.org/unsd/methodology/m49 (accessed on 20 June 2022) Eastern Europe: Belarus, Bulgaria, Czechia, Hungary, Poland, Republic of Moldova, Romania, Russian Federation, Slovakia, Ukraine. Northern Europe: Denmark, Estonia, Finland, Iceland, Ireland, Latvia, Lithuania, Norway, Sweden, the UK of Great Britain and Northern Ireland. Southern Europe: Albania, Bosnia and Herzegovina, Croatia, Greece, Italy, Malta, Montenegro, North Macedonia, Portugal, San Marino, Serbia, Slovenia, Spain. Western Europe: Austria, Belgium, France, Germany, Liechtenstein, Luxemburg, Monaco, Netherlands, and Switzerland). 

Well-being is constructed through two questions included in the survey about “Life satisfaction” and “Happiness,” rated on a 1–10 scale to assess “How satisfied are you with your life these days?” and “How happy would you say you are?” 

The activity type they engaged in is made up of two variables: “Started working from home as a result of the COVID-19 situation” (dichotomic) and “Employment status” (employee, self-employed without and with employees), creating a six-range variable. 

Finally, the “Feelings about work–life balance” is a five-category variable through which respondents scored four items asked by the survey: “Kept worrying about work when you were not working”, “Felt too tired after work to do some of the household jobs which needed to be done”, “Found that your job prevented you from giving the time you wanted to your family”, “Found it difficult to concentrate on your job because of your family responsibilities”, and “Found that your family responsibilities prevented you from giving the time you should to your job”.

## 3. Results

Findings are presented in three sections. The first section aims to capture workers’ variation in responses by gender due to the lockdown situation in the COVID-19 pandemic, according to self-reported life satisfaction. The second section addresses men’s and women’s well-being by geographical areas, which may be related to the gender-regime tradition, public policy, social benefits, and labor market characteristics. The third section attempts to discover the well-being and worries of men and women from spending more time than usual at home. 

### 3.1. Life Satisfaction and Gender of Participants, General Overview

The analysis finds an association between the life satisfaction of workers and the gender of respondents (0.000 F = 34.01), where men have higher life satisfaction than women. They appear to be affected differently by the crisis and the situations both of working at home and taking care of family. While women’s well-being is associated with work type (average life satisfaction for women teleworking is 6.90 and those who remain in the workplace 6.18), while men’s well-being is related to geographic region. 

The classification tree analysis reveals that both men and women teleworking have higher satisfaction levels than those who remained in the workplace, making teleworking seem like an optimal solution to uphold performance in times of social acceleration. However, female responses suggest worries about their daily experience as front-line workers, while men’s responses seem to reflect a perception of health risk. Northern countries show the lowest mortality rates, and male life satisfaction reaches 7.25, while higher excess mortality rates correspond to Southern countries, where men report the lowest life satisfaction, 6.21 (data on mortality rates are available at https://ec.europa.eu/eurostat/cache/digpub/regions/#weekly-deaths (accessed on 20 June 2022)). 

Results point to risk perception being the baseline for life satisfaction, where men are more likely to value staying safe at home, and women who are not teleworkers rate the lowest life satisfaction (6.18). Women’s worries seem to hinge on work, while the health situation appears more concerning for male respondents. This result suggests that segregation of the labor market matters to women, who expressed concerns about the possibility that working remotely would hinder their job performance. 

### 3.2. Life Satisfaction and Happiness by Gender Regime Broken down by European Regions

By regions, workers from Southern countries who do not telework show minimum life satisfaction (5.82 for women from Southern countries not teleworking and 6.03 for men with the same features), while male teleworkers from Northern countries and female teleworkers display the highest levels of life satisfaction (7.25 for Northern men teleworkers and 7.09 among women teleworkers from Western countries).

The gender gap is clear according to these results (Table 2). Men show higher self-reported life satisfaction than women in almost all regions, except Southern countries, where women show a slightly higher perception of life satisfaction than men. Southern European countries present the lowest life satisfaction ratio, while Northern and Western countries have the highest.

To learn more about workers’ experiences of spending more time at home than usual, we compared the life satisfaction of male and female teleworkers during the pandemic. In line with previous findings, we expected differences at least in Southern countries, leading us to search for likely variation by employment status (Figure 1).

The analysis finds higher rates of male than female satisfaction in Eastern and Western countries. Data may suggest that men found working at home satisfactory, with scores around 7. Gender differences are revealed for self-employed without employees (freelancers) in Northern countries and employees and freelancers in Southern countries, where women report higher life satisfaction than men.

### 3.3. Spending Time at Home, Well-Being, and Worries

In July, the second round of the survey included relevant questions for our aims on caring and household work during the last month. Results reported by Eurofound [25] reveal persistent gender differences, where women were generally more involved in spending time caring for family and doing household work (35 h/week caring and 18 h/week housekeeping and men spent 25 h/week caring for family members and 12 h/week housekeeping). Therefore, considering the whole population, it concludes that lockdown did not entail a significant transformation of gender relations or transformation of traditional gender-role distribution at home.

Still, over one-third of respondents worked fewer hours during the pandemic, where the decreased hours were more usually for men (−4.9%) than women (−5.2%), which significantly changes the scenario for male workers. It raises a question about the daily home lives of male teleworkers: How did this “spare time”—since they worked fewer hours and did not transform their roles at home—affect their well-being? We wondered if it would interfere with men’s self-reported worries about life–work balance, at least, because they were exposed to spending more time in family care and housekeeping. If they did take on care responsibilities to a lesser degree than women, did men perceive this time as more relaxed than women who must juggle the work–life balance more?

We look at the distance between happiness and life satisfaction to estimate what the subjectivation is of their quality of life during this time. As Figure 2 displays, they reported that they “always” had difficulties concentrating on their jobs because of family and “never” felt too tired after work to do some of the household jobs that needed to be done (Figure 2). According to the first indicator, they probably felt overwhelmed by the family around them while working, and according to the second, they did not do so much housekeeping that they felt tired. Jointly, this points to them keeping their routines for work and domestic duties unchanged.

Data suggest that women continue with all gender roles despite lockdown, coping with work–life balance in the new situation. Delving into the impact of gender on well-being, we compare men’s and women’s self-reported life satisfaction. According to Table 3 below, women felt too tired “sometimes” and “most of the time” more than men, but women self-reported higher proportional life satisfaction than men. Findings disclose a similar trend for the following items. Women found it more difficult to concentrate on their jobs because of family in comparison to men (and in accordance with gender-role distribution), but women reported proportionally more life satisfaction than men. Men and women felt worries about work, even though men are less centered on household and care responsibilities, and their life satisfaction ratios were close, and sometimes even lower, than that of women (for instance, “always” felt too tired: 4.26 men; 5.72 women). This suggests that women exhibit more adaptable and resilient attitudes, while men who made very few changes to their habits involving work–life balance felt life satisfaction as low as women.

## 4. Discussion

According to data on teleworking during the pandemic, this work mode is not related to advances in greater gender equality [1,2]. Remote work is insufficient if the labor market and gender regimes remain biased [10]. If preceding findings highlighted that teleworking had a limited impact on women advancing in the labor market and the public domain, these results also suggest it is a poor mechanism for advancing toward a more equal gender regime in the domestic arena. Some women may find advantages in teleworking because of time management and the self-organization of tasks corresponding to work–life balance in the household [5,15]. On the contrary, men spending more time at home involves almost no impact on a more equal gender regime with equal gender-role distribution. If women have taken the risk of tackling work–life balance as they advance in joining the labor market, men do not have the risk of taking more responsibility for care and household work when circumstances place them at home.

In the career-centric debate [12], women appear to be focused on careers without neglecting family care responsibilities. On the contrary, men seem to keep traditional gender privileges as breadwinners, even if the pandemic may have caused a social change, keeping them at home. Although the pandemic could have pushed men toward making social changes, they did not change. A possible shrinking gender gap points to there not being enough time caring for the family and household to modify the gender regime [19]. In that sense, the gender regime remains unbroken.

Since men do not take the risk of tackling the lion’s share of the work–life balance, they self-reported a higher proportion of life satisfaction than women. While women’s concerns are related to continuing presence at the workplace or, conversely, having the opportunity to telework, men’s self-reported satisfaction depends on their geographic region (probably health risk situation). Southern countries are both seriously affected by the pandemic and by more domestic division of work, reporting poor life satisfaction [25]. As Table 2 shows, Southern countries reveal the lowest life satisfaction, whereas Northern countries and Western countries have the highest rates. However, a general overview of the data discloses that keeping traditional roles entails serious risk for women; men from Southern countries display the lowest life satisfaction (surpassing women from the same region), which may point to pandemic worries and socio-economic factors, jointly with gender issues. Presenteeism and teleworking previous to the pandemic culture in different regions might also have raised the dissatisfaction of men in Southern countries. Accordingly, women appear more satisfied with teleworking when they have the opportunity with this work arrangement. The employment status of women is also relevant to learning more about differences between European regions. Women freelancers (self-employed without employees) who started to work from home as a result of the pandemic in Northern and Southern European countries display higher satisfaction compared to male freelancers [8]. The pandemic would seem to have given them the opportunity to do remote work and be accepted by the labor market while taking advantage of organizing work–life time.

Juggling domestic and labor responsibilities during the lockdown, women, in due course, felt tired and worried about work when they were not working and had difficulties concentrating because of family needs [15,18,19]. Women felt their well-being deteriorated, but surprisingly, men also indicated poor well-being, at least stronger in proportion than women who take on more care responsibilities. Although Eurofound [25] pointed out women’s low resilience, we think that these findings display toughness to withstand adverse conditions working at home, more than men. Female low resilience would relate to socio-economic issues, but women would lead with a better work–life balance at home than men. With regard to men, the distance between happiness and life satisfaction suggests few endeavors to transform gender relations and a proportional feeling of quality of life.

The quality-of-life assessment is key for undertaking minimal social changes. It suggests that it is very unlikely that gender transformation can be forced on men in the domestic arena. However, we frankly think that disseminating results—such as this work—may illuminate and raise men’s awareness of the need to transform inequality at home to improve family life beyond the private habitus. Daily routines are rooted in the androcentric status quo, keeping the same roles, or with minimal changes, that reinforce men’s roles having a privileged status. Social and technological changes such as public policy and normative measures are putting cracks into gender regimes [15,20], and now may be the time to show men how unequal the situation is and trigger their involvement in the household.

Further research is needed to discover the determinants of well-being for male and female workers regarding work–life balance. Health status and poor labor conditions are also key variables to determine different responses. Teleworking and social acceleration are components of neoliberalism, and workers are increasingly embroiled in subtle and strong cultural variables, making us need to learn more about men’s and women’s awareness and resistances, depending on their lifestyles and socio-economic contexts. The heterogeneity of workers depending on labor activity, working hours, type of family, and household, among others, blurred our evaluation of the influence of telework on population groups. We need a more in-depth investigation into factors that separate groups of workers, including an eco-social framework of analysis that would contribute to explaining differences.

## 5. Conclusions

This research addresses the impact of male and female workers’ well-being during the COVID-19 lockdown, where spending time at home is relevant to determine any social changes to the gender regime. Men appear to have greater life satisfaction than women. The source of worries for men is keeping safe at home, dependent on the general rates of health risk in their countries, while women’s concerns depend on the availability of teleworking. Women workers present a work-centric orientation, although further analysis also proves they are in charge of family care and housekeeping.

By regions, men self-reported high life satisfaction in almost all areas. Surprisingly, in Southern Europe, where all studies underline strong domestic division and work–life conflict, women’s perception of life satisfaction during the lockdown is slightly higher than that of men. However, Southern men and women together display the lowest life satisfaction in Europe. Interestingly, women self-employed without employees in Northern and Southern Europe report a higher proportion of life satisfaction than men.

Although evidence shows that men made few social changes regarding spending more time at home doing care and housekeeping, we want to know more about men’s experiences at home. Findings report high difficulties concentrating on their jobs because of family and, correspondingly, men’s minor involvement in work–life balance, but they “never” felt too tired after work to do housekeeping. Women self-reported that they were “too tired,” had difficulties concentrating on their jobs, and worried about work when they were not working, but compared to men, they felt life satisfaction to a higher degree than would be expected.

These findings point to men’s high resistance to taking risks of spending more time doing care and housekeeping even during the pandemic and spending more time at home. Expected changes in gender regime seem remote, even though teleworking is widespread and equally affects men and women. We suggest twofold measures: providing men with incentives and messages about gender inequities at home; and social and economic support for women as the main players in work–life balance, even in special circumstances.

## Figures and Tables

**Figure 1 ijerph-19-08468-f001:**
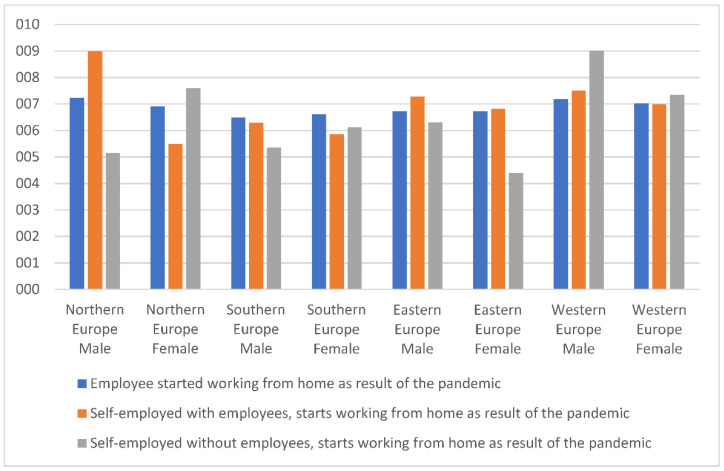
Life satisfaction of men and women teleworkers, employment status across European regions. Source: Own elaboration based on Eurofound data.

**Figure 2 ijerph-19-08468-f002:**
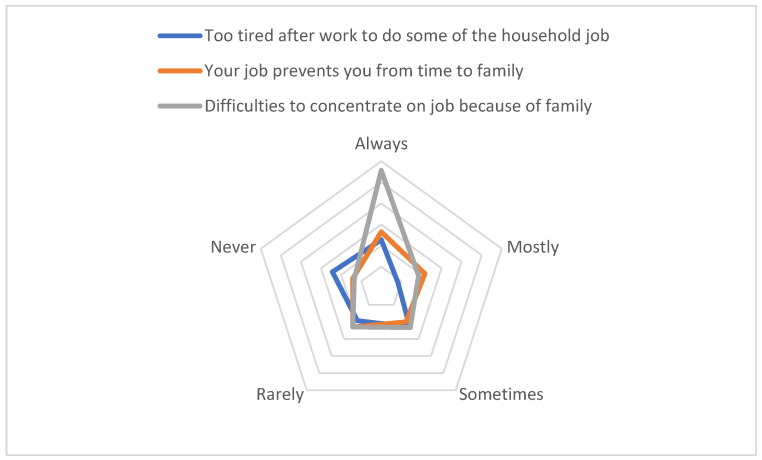
Male distance between life satisfaction and happiness. Source: Own elaboration based on Eurofound data.

**Table 1 ijerph-19-08468-t001:** Characteristics of the people who make up the study sample (*n* = 92,269).

		Valid Percentage
Round	One (April–May 2020)	73.9%
	Two (June–July 2020)	26.1%
Gender	Male	47.4%
	Female	52.6%
Age group	18–34	24.2%
	35–49	24.4%
	50+	51.4%
Employment status	Employee	44.9%
	Self-employed with employees	2.3%
	Self-employed without employees	5.9%
	Unemployed	7.6%
	Unable to work due to long-term illness or disability	3.1%
	Retired	25.2%
	Full-time homemaker/fulfilling domestic tasks	3.8%
	Student	7.2%
European region	Northern Europe	7.1%
	Southern Europe	30.9%
	Eastern Europe	20.2%
	Western Europe	41.8%
Household size	1	19.9%
	2	37.9%
	3	19.4%
	4	15.3%
	5	5.1%
	6+	2.5%

Source: Own elaboration.

**Table 2 ijerph-19-08468-t002:** Life satisfaction of men and women in European regions.

Northern Countries		Southern Countries		Eastern Countries		Western Countries	
Male	6.84	Male	6.07	Male	6.33	Male	6.75
Female	6.74	Female	6.12	Female	6.19	Female	6.44
Total	6.78	Total	6.10	Total	6.25	Total	6.59

Source: Own elaboration based on Eurofound data.

**Table 3 ijerph-19-08468-t003:** Worries and life satisfaction of men and women.

		Male	Life Satisfaction	Female	Life Satisfaction
Felt too tired after work to do household tasks	Always	4.2%	4.26	4.8%	5.72
	Most of the time	18.5%	6.04	20.1%	6.09
	Sometimes	38.4%	6.68	43.5%	6.65
	Rarely	23.0%	6.81	19.8%	6.84
	Never	15.9%	6.97	11.8%	7.16
Difficult to concentrate on job because of family	Always	1.1%	4.80	2.8%	5.96
	Most of the time	7.9%	5.98	10.1%	6.11
	Sometimes	31.7%	6.41	30.6%	6.46
	Rarely	35.1%	6.75	30.8%	6.71
	Never	24.3%	6.75	25.6%	6.86
Worried about work when not working	Always	8.0%	5.42	9.3%	6.12
	Most of the time	29.2%	6.16	27.4%	6.32
	Sometimes	37.3%	6.70	34.5%	6.61
	Rarely	15.4%	6.87	16.7%	6.96
	Never	10.0%	7.26	12.0%	7.01

Source: Own elaboration based on Eurofound data.

## Data Availability

Restrictions apply to the availability of these data. Data was obtained from Eurofound with the permission of Eurofound through a data analysis request.

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
