# Peer review of "Work–Life Balance and Teleworking: Lessons Learned during the Pandemic on Gender Role Transformation and Self-Reported Well-Being"

_ijerph, 2022, doi:10.3390/ijerph19148468_

Round 1

Reviewer 1 Report

The issues raised in this paper are important and are of concern to everyone in the world that is concerned by the social impacts of the pandemic.  To this end, you might have acknowledged that impacts were particularly severe for women and for other marginalized groups.  You might also tighten up your use of the terms "sex" and "gender", as one does not imply the other. For example, what do you mean by women's roles?  This is a gender (performativity, expectations, cultural, etc.) issue rather than a biological one.  Also, are you talking about equality, or equity?  The two are quite different.  

In your introduction, I am not sure what you mean by "flagship". Perhaps you are referring to the significance of the problem of inequities in the teleworking world (for example, digital access)? In this section, the reader would benefit from a broader discussion about the challenges - social, cultural, economic, political - of teleworking in general, not just in crisis situations.

As a qualitative researcher, I am unfamiliar with the methods used and suspect others are in the same situation . Will you please provide a more clear definition? Also, please provide more information on the survey (which is little known in North America) and define the various "Europes". For example, I was not sure what Southern Europe comprised.  This discussion could be strengthened with a discussion of cultural differences in gender role expectations as well.

Author Response

Thank you very much for your suggestions, we have reviewed and made changes to the manuscript in line with your comments. We would like to express our thanks for these fruitful insights that will help to improve the language and content of the article. A more detailed list of changes is presented below:

First, the introduction has been changed, stressing the contradictory expectations of the effects of working at home on women's work-life balance, also in line with reviewer number 2. We also expanded upon the discussion about the socio-cultural and economic challenges involved in teleworking

Second, we reviewed the terms “sex” and “gender”; we checked the survey methodology and the questions asked in the survey that mention gender, not sex. In addition, we reviewed “equality” and “equity” and “flagship”. We hope the new wording makes the manuscript clearer.

Third, we include more information about the survey and the analysis. More information about which European countries are in each region has been included in the manuscript. And, in accordance with both reviewers, more information about cultural differences in gender role expectations have been added throughout the manuscript.  

Reviewer 2 Report

In the introduction the contradictory expectations of the effects of working at home for women's work life balance could be more stressed. The discussion does tackle gender roles but not enough, in my view, the prescritive dimension of women's household and caring roles which is transversal to northern and southern European countries.

Even if EUROFOUND is a well known regular survey some more information about the sample is lacking to sustain these specific results: number of participants, age and family status of the respondents - living alone, living in couple, for example.

Author Response

Thank you very much for your valuable suggestions. Your comments, jointly with reviewer 1, have improved the manuscript. Please see below a short list of changes made to the manuscript.

  1. We changed the introduction section in line with your comments
  2. We added the suggested discussion on prescriptive gender roles in Northern and Southern European countries.
  3. In methodology, we include more detailed information on the survey.

Round 2

Reviewer 1 Report

Thank you for your considered revisions of this paper.  Much improved!  I think, however, that you still need to look at your use of "women" and "gender".  Are you talking only about women?  If so, use that term.  Talking about gender suggests your use of social constructs - such as the gender pay gap, or gender roles.